# Obesity Contributes to Transformation of Myometrial Stem-Cell Niche to Leiomyoma via Inducing Oxidative Stress, DNA Damage, Proliferation, and Extracellular Matrix Deposition

**DOI:** 10.3390/genes14081625

**Published:** 2023-08-15

**Authors:** Sadia Afrin, Gregory W. Kirschen, Mostafa A. Borahay

**Affiliations:** Department of Gynecology and Obstetrics, The Johns Hopkins Hospital, Baltimore, MD 21287, USA; dolla.bihs@gmail.com (S.A.); gkirsch7@jh.edu (G.W.K.)

**Keywords:** adipocytes, adipokines, leptin, fibroid, collagen

## Abstract

Leiomyomas (fibroids) are monoclonal tumors in which myometrial stem cells (MSCs) turn tumorigenic after mutation, abnormal methylation, or aberrant signaling. Several factors contribute to metabolic dysfunction in obesity, including abnormal cellular proliferation, oxidative stress, and DNA damage. The present study aims to determine how adipocytes and adipocyte-secreted factors affect changes in MSCs in a manner that promotes the growth of uterine leiomyomas. Myometrial stem cells were isolated from the uteri of patients by fluorescence-activated cell sorting (FACS) using CD44/Stro1 antibodies. Enzyme-linked immunosorbent assay (ELISA), Western blot, and immunocytochemistry assays were performed on human adipocytes (SW872) co-cultured with MSCs and treated with leptin or adiponectin to examine the effects of proliferation, extracellular matrix (ECM) deposition, oxidative damage, and DNA damage. Co-culture with SW872 increased MSC proliferation compared to MSC culture alone, according to 3-(4,5-dimethylthiazol-2-yl)-2,5-diphenyltetrazolium bromide (MTT) results. The expressions of PCNA and COL1A increased significantly with SW872 co-culture. In addition, the expression of these markers was increased after leptin treatment and decreased after adiponectin treatment in MSCs. The Wnt/β-catenin and TGF-β/SMAD signaling pathways promote proliferation and ECM deposition in uterine leiomyomas. The expression of Wnt4, β-catenin, TGFβ3, and pSMAD2/3 of MSCs was increased when co-cultured with adipocytes. We found that the co-culture of MSCs with adipocytes resulted in increased NOX4 expression, reactive oxygen species production, and γ-H2AX expression. Leptin acts by binding to its receptor (LEP-R), leading to signal transduction, resulting in the transcription of genes involved in cellular proliferation, angiogenesis, and glycolysis. In MSCs, co-culture with adipocytes increased the expression of LEP-R, pSTAT3/STAT3, and pERK1/2/ERK/12. Based on the above results, we suggest that obesity may mediate MSC initiation of tumorigenesis, resulting in leiomyomas.

## 1. Introduction

Uterine leiomyomas (fibroids) are prevalent, benign smooth muscle cell tumors of the muscular layer of the uterus, which can result in serious morbidity for those affected, including abnormal uterine bleeding, infertility, and pelvic pain [1]. Emerging evidence suggests that cardiometabolic risk factors, including hypertension, dyslipidemia, and obesity may be associated with fibroid development [2,3,4]. For instance, a meta-analysis of 24 studies including over 325,000 participants identified a positive relationship between body mass index (BMI) and fibroid prevalence, with a statistically significant odds ratio (OR) of 1.19 [3]. In another study, a direct association between BMI and uterine fibroid size was observed, with a 7.56 g increase in uterine weight for every 1-point increase in BMI [5].

The mechanisms by which obesity may be causally linked to fibroid development have been increasingly elucidated in recent years. Adipokines, adipocyte-released signaling molecules, may promote fibroid tumor proliferation via inflammatory pathways that are upregulated in obese patients [6]. Among patients with fibroids, pro-inflammatory cytokines and the neutrophil–lymphocyte ratio have been shown to correlate positively with triglyceride levels, suggesting that dyslipidemia, known to induce a pro-inflammatory milieu, may contribute to fibroid growth and development [7]. The local oxidative stress that accompanies the obese state may further drive fibroid growth via genomic oxidation at the *MED12* locus, which is fueled by NOX4 and TGF-β3 signaling [8,9].

Various genetic pathways have demonstrated relevance in fibroid biology. Transcriptomics analysis of the fibroids, normal myometrium or endometrium of fibroid uteri have revealed alterations in microRNA (miRNA) as well as an upregulation of *E-Rcβ*, *MMP1*, and *TIMP-2* that regulate growth and inflammatory cellular cascades [10,11]. Relatedly, transforming growth factor-β (TGF-β) signaling, induced by the hypoxic microenvironment of fibroids, has been implicated in fibroid growth via NOX4-mediated oxidative stress and the generation of ROS [8,12]. Given the pro-inflammatory milieu induced by obesity, we hypothesized that an adipocyte-rich environment, which has been associated with alterations in miRNAs matrix metalloproteinases and increased ROS, would provide such a catalyst for fibroid growth [13,14]. 

A class of adipocyte-derived molecules known as adipokines affect a host of physiological processes. As an endocrine organ, adipocytes secrete a variety of factors including what are known as adipokines, which are hormonally and immunologically active [15]. Among their identified roles, adipokines such as leptin are responsible for regulatory T cell modulation, and adiponectin regulates endothelial-dependent vasodilation [16,17,18].

Despite these advances, the precise influence of adipocyte-derived factors on myometrial cells that affect fibroid formation and propagation remain unknown. We hypothesized that adipokines interacting with myometrial stem cells (MSCs), the precursor cells of fibroids, would activate inflammatory pathways. Using a novel human-derived MSC-adipocyte co-culture system, we dissect the complex endocrinological and immunological interactions between these two cell types that provides insights into the causal relation between obesity and fibroid development.

## 2. Materials and Methods

### 2.1. Isolation and Culture of Myometrial Stem Cells

MSCs were isolated as previously described [19]. Briefly, cells were isolated from patients undergoing hysterectomy at the Johns Hopkins University Hospital. The Institutional Review Board of the Johns Hopkins University reviewed and approved the study (IRB00196175), and informed consent was obtained from all patients to participate. Tissue was transferred to the laboratory immediately after surgery and washed several times with a Hanks’ Balanced Salt Solution (HBSS, Thermo Fisher Scientific, Waltham, MA, USA) without calcium or magnesium. Tissue was then manually cut into 1–2 mm^3^ specimens and incubated in sterile HBSS (without calcium, or magnesium) with collagenase (Worthington, Lakewood, NJ, USA), deoxyribonuclease (DNase, Sigma-Aldrich, St. Louis, MO, USA), antibiotic–antimycotic mixture (Thermo Fisher Scientific), and HEPES buffer solution (Thermo Fisher Scientific) at 37 °C on a shaker for 4–8 h. The digest was filtered through a 100 µm and then a 40 µm filter and cultured in Dulbecco’s Modified Eagle Medium/Nutrient Mixture F12 (DMEM/F-12) (Thermo Fisher Scientific) medium supplemented with HEPES, L-glutamine, 10% FBS, and 1% antibiotic–antimycotic.

### 2.2. Preparation of MSC-Adipocyte Co-Culture and Leptin and Adiponectin Treatments

The human adipocyte SW872 cell line was obtained from the American Type Culture Collection (ATCC; Manassas, VA, USA) and maintained in culture conditions of DMEM/F12 with 10% FBS and 1% antibiotic–antimycotic in 5% CO_2_ at 37 °C. Human adipocyte (SW872) cells were co-cultured with MSCs [6]. A Transwell system was used consisting of a 0.4 mm porous membranes from Corning (Corning, New York, NY, USA) to co-culture the cells with their respective media. We cultured the Transwell system for eight days and changed 50% of the medium every 48 h. We then harvested the cells. For the leptin treatment condition, we treated the cells with 100 ng/mL leptin for 48 h. After 48 h of incubation, the cells were harvested for protein quantification and analysis. For adiponectin treatment, MSCs were treated with 20 μg/mL of adiponectin for a total of 48 h. After 48 h of incubation, the cells were harvested for protein quantification and further analysis.

### 2.3. Protein Extraction and Western Blot Analysis

Protein extraction and Western blot analysis were performed as previously described [6,20]. Following co-culture and leptin or adiponectin treatment, we harvested and lysed the MSCs in an ice-cold lysis buffer (radioimmunoprecipitation assay buffer; MilliporeSigma, Burlington, MA, USA). The buffer was composed of a phosphatase and protease inhibitor cocktail (MilliporeSigma). We combined the same amounts of protein lysates with 4–12% Bis-Tris gradient gels (Thermo Fisher Scientific) and transferred them onto nitrocellulose membranes (Thermo Fisher Scientific). For 1 h at room temperature, we blocked the membranes by soaking them in Tris-buffered saline with 0.1% Tween-20 (TBST; Thermo Fisher Scientific) in 5% nonfat milk. The membranes were then incubated for 24 h with the following specific primary antibodies: PCNA (CST, #13110), COL1A (Thermofisher, PA5-29569), Wnt-4 (Invitrogen, 701857), β-catenin (CST, #8480), TGF-β3 (Invitrogen, PA5-32630), pSMAD2/3 (CST, #8828), LEP-R (Invitrogen, PA1-053), anti-phosphorylated ERK1/2 (CST, #4370S), anti-total ERK1/2 (CST, #4695S), anti-total STAT3 (CST, #9139S), anti-phosphorylated STAT3 (CST, #9145S), NOX4 (abcam, ab133303), γH2AX (CST, #2577) and β-actin (Sigma, A3854). These membranes were left overnight at 4 °C on a rocker and diluted in 5% BSA (1:1000). We then washed the membranes with TBST and incubated the membranes with the corresponding horseradish peroxidase (HRP)-conjugated secondary antibodies for 1 h at room temperature. Using an Azure Imager c300 (Azure Biosystems, Dublin, CA, USA), we visualized the membranes. The protein band signals were quantified using the NIH ImageJ software (version 1.52r) [21].

### 2.4. MTT Analysis

Cell viability analysis was performed as previously described [19]. Briefly, MSCs were plated onto a 24-well plate and grown overnight. Cells were co-cultured with adipocytes as described above, or with leptin or adiponectin treatment, as described above. Viability was assessed by (3-[4,5-dimethylthiazol-2-yl]-2,5-diphenyltetrazolium bromide) (MTT) staining by measuring absorbance at 500–600 nm.

### 2.5. Reactive Oxygen Species (ROS) Production Assay

We performed an 8-day co-culture of adipocytes and MSCs, as described above. As we previously described, changes in intracellular ROS levels were determined using the fluorogenic probe 5-(and-6)-chloromethyl-2′,7′-dichlorodihydrofluorescein diacetate acetyl ester (CM-H2DCF-DA; Invitrogen, Eugene, OR, USA) [8]. After 8 days, co-cultured cells were treated with 5 μM CM-H2DCF-DA added to each well, and cells were incubated at 37 °C for 30 min. Cells were then washed with HBSS. Empty wells without plated cells were used as a negative control. Changes in DCF fluorescence were recorded on a microplate reader (CLARIOstar, (BMG LabTech, Cary, NC, USA) at 485 nm excitation and 528 nm emission. Results are expressed as fold change or arbitrarily in fluorescence units (FU). Experiments were performed in quintuplicate.

### 2.6. Statistical Analysis

Data were analyzed using GraphPad Prism version 6.01 for Windows (GraphPad (Insight Partners, Graphpad Holdings, LLC, Boston, MA, USA). A D’Agostino–Pearson omnibus normality test was used for data distribution. Unpaired Student’s *t*-test and one-way ANOVA with an appropriate Tukey’s post hoc test were used according to the type of experiment. The nonparametric Mann–Whitney U and Kruskal–Wallis followed by Dunn’s test were used for mRNA and protein expression levels, respectively. All experiments were performed in duplicate and repeated three independent times from different patients. Error bars represent a standard error of the mean (SEM). *p*-values < 0.05 were considered statistically significant (* denotes *p* < 0.05; ** denotes *p* < 0.01).

## 3. Results

Adipocytes induce ROS production and activate NOX4 and γH2AX expression in myometrial stem cells co-culture.

To dissect the relationship between adipocytes and MSCs, we established a co-culture system in which human-derived MSCs taken from patient hysterectomy samples were cultured with human adipocyte (SW872) cells. In vivo, obesity and excessive adiposity are known to increase oxidative stress through the generation of free fatty acids (FFAs) and hyperglycemia [22]. NAPH oxidase 4 (NOX4) is a potent hydrogen peroxide producer that is typically used for cellular defense against phagocytosed pathogens but which becomes aberrantly upregulated in the obese state [23,24]. Histone γH2AX is an important actor in the DNA damage response, alerting the cell to areas of DNA damage to target repair [25]. We hypothesized that adipocyte co-culture would result in increased free radical production with an increase in NOX4 and compensatory γH2AX upregulation (ostensibly in response to DNA damage). As predicted, we observed increased ROS generation, increased NOX4, and increased γH2AX in MSCs co-cultured with adipocytes as compared to MSCs cultured alone (Figure 1A–C). 

### 3.1. Activation of Leptin Receptor in Myometrial Stem Cells Co-Culture with Adipocyte

We reasoned that MSCs would be responsive to adipocyte-derived cues such as the hormone leptin and have leptin receptors (LEP-Rs). Leptin is an adipokine secreted in proportion to the amount of adipose tissue present in an organism, and its known functions include signaling to the central nervous system to indicate sufficient energy levels to decrease the drive to consume calories [26]. Moreover, leptin resistance is observed in the obese state [26]. Aside from signaling energy balance, however, leptin has been implicated in inflammation and autoimmunity [27,28].

We hypothesized that adipocytes would upregulate leptin receptor expression and coupled receptors/second messengers. We co-cultured MSCs with adipocytes and indeed observed an upregulated expression of LEP-R in co-cultured versus singly cultured MSCs (Figure 2A). In tandem, we observed increased pSTAT3/total STAT3 and increased pERK1-2/total ERK1/2 expression in co-cultured MSCs as compared to singly cultured MSCs, suggesting that the STAT3 and ERK1/2 pathways, important for cellular growth and proliferation, are upregulated in the presence of adipocytes (Figure 2B,C).

### 3.2. Activation of TGF-β3/SMAD2 and Wnt4/B-Catenin Pathway in Myometrial Stem Cells Co-Culture with Adipocyte

Having observed altered MSC signaling related to cell growth, in the presence of adipocytes and adipocyte-secreted factors, we next interrogated the cellular mechanisms underlying this phenomenon. We hypothesized that the Wnt/β-catenin and TGF-β/SMAD signaling pathways, which promote proliferation and ECM deposition in fibroids, would be upregulated in the presence of adipocytes [19]. We therefore co-cultured MSCs and adipocytes and quantified the protein expression of Wnt4, β-catenin, and TGF-β3. As anticipated, we noted a significantly elevated expression of these proliferation and growth-related factors in MSCs co-cultured with adipocytes as compared to MSCs cultured alone (Figure 3A–C), and a commensurate increase in phosphorylated SMAD2/3, the downstream signaling partner of TGF-β3, as observed immunohistochemically (Figure 3D).

### 3.3. Adipocytes Co-Culture and Adipokine Treatment Increases Myometrial Stem Cells Proliferations and Type 1 Collagen Production

Cellular proliferation and extracellular membrane (ECM) deposition are required for leiomyoma development. We performed a proliferation assay to determine whether adipocyte co-culture would prompt the proliferation of MSCs. We found that cell viability was significantly increased in the co-culture condition as compared to MSC conditioned alone, which was paralleled by significantly increased PCNA (proliferative marker) and COL1A (collagen production) protein expression (Figure 4A). Adipocytes produce and secrete various small signaling molecules to regulate the endocrine system. Leptin is a peptide hormone synthesized by adipocytes that acts in the hypothalamus to stimulate appetite and promoting positive energy balance in addition to roles in growth, inflammation, angiogenesis, and lipolysis [29]. Another important adipocyte-derived factor is adiponectin, which is a signaling protein implicated in insulin resistance, cardiovascular disease, and metabolic syndrome [30]. Of note, adiponectin levels are inversely correlated with insulin resistance and thus tend to be low in obese patients and those with the metabolic syndrome [30]. We hypothesized that adipocytes increase MSC proliferation via endocrine/paracrine signaling using leptin and adiponectin. We therefore cultured MSCs in the presence of supra-physiological levels of leptin and adiponectin. As expected, we observed increased PCNA and COL1A expression in MSCs cultured with leptin and lower expression of these markers in the presence of adiponectin, suggesting that high leptin and low adiponectin, as observed in the obese state, promote MSC proliferation and collagen deposition (Figure 4B,C).

Altogether, our findings demonstrate that adipocytes directly influence MSC behavior, upregulating their proliferation and collagen production. We further describe a novel mechanism by which obesity may directly promote fibroid generation involving endocrine signaling and ROS generation. Figure 5 displays a diagram depicting how adipose tissue influences fibroid tumor generation and growth via the above mechanism. We propose that adipocytes increase pro-inflammatory cytokines, adipokines, and other endocrine signals to activate NOX4, promoting the generation of intracellular ROS, leading to DNA damage, a DNA damage response, DNA mutation and subsequent fibroid tumor development and growth.

## 4. Discussion

Uterine fibroids are increasingly recognized as a cardiometabolic disease marker and sequela. High BMI and the metabolic syndrome have consistently been associated with fibroid propensity in several epidemiological case-control studies [3,31,32]. Until this point, however, a clear causal link between obesity and fibroid generation has never been convincingly demonstrated. Several hypotheses have been put forth to attempt to explain this apparent clinical association. For instance, visceral fat accumulation that occurs in the obese state changes electrical impedance, which may influence fibroid growth [31,33]. We and others have proposed an inflammatory hypothesis, according to which local chronic inflammation triggers a series of intracellular cascades within myometrial smooth muscle cells or stem cells, leading to increased Wnt/β-catenin and increased NF-κB and TNF- ⍺, with resultant hyperproliferation and extracellular matrix deposition (ECM) [34,35]. Indeed, there is evidence that each of these signaling molecules are upregulated in fibroids, along with others including activin (activates fibroblasts to synthesize ECM, TGF-β (activates inflammation and tumorigenesis), CXCL12 (recruits bone-marrow-derived stem cells) and miRNA-200c (disinhibits NF-κB production) [34,36,37,38,39]. 

Several provocative studies have suggested that there may be a direct link between obesity, the metabolic syndrome, and fibroids. Obesity is associated with characteristic physiological changes, including imbalances of energy regulation hormones, adipokines such as leptin and adiponectin and increased levels of estrone [40,41]. We previously found that leptin promotes human leiomyoma cell proliferation via the JAK/STAT3 and MAPK/ERK pathways.

Performing cardiometabolic profiling of patients with fibroids, we found that compared to controls without fibroids, those with fibroids were more likely to be obese and have the metabolic syndrome, essential hypertension, diabetes, and dyslipidemia, with the pro-inflammatory milieu and diet serving as likely strong mediators [34,42,43,44]. Some or all of these mechanisms may be at play to explain the link between obesity and subsequent fibroid formation [45].

Aside from the direct effects of obesity on MSCs and fibroid cells, there is emerging evidence that other features of the metabolic syndrome may contribute to fibroid development. For example, we and others have found that fibroids are associated with essential hypertension [4,46,47,48]. Whether the metabolic milieu unilaterally influences fibroid growth or whether a bidirectional feedback loop exists wherein fibroids may actually contribute to hypertension/metabolic syndrome remains to be explored. We recently found that the surgical removal of fibroids was associated with a modest decrease in systolic blood pressure [46]. Others have found that hypertension treatment, either with angiotensin-converting enzyme inhibitors (ACE-i) or β blockers, was associated with a reduced incidence of fibroids among women with hypertension [48,49]. Thus, the possibility of bidirectional signaling between fibroids and hypertension (or other cardiometabolic risk factors) may exist. The current work sought not to focus on the interplay between the systemic vasculature and fibroid pathogenesis but rather the complex endocrinological signaling between adipocytes and MSCs.

Furthermore, excessive adipose tissue aromatizes androstenedione to estrone, which is an estrogen that may influence fibroid growth [50]. Aromatase inhibitors have been used to treat fibroids by blocking estrogen production, leading to a clinically meaningful decrease in fibroid size [51]. Thus, it stands to reason that this is another mechanism by which adipose tissue may influence fibroid growth. Estrogen contributes to fibroid pathogenesis (and treatment of fibroids) in complex ways. Fibroid growth is estrogen-dependent, explaining their tendency to appear after onset of puberty and regress after the menopausal transition [52]. 17β-estradiol (E_2_), targeting nuclear estrogen receptor ⍺ (ER⍺), induces fibroid cell proliferation under physiological (cyclical and at physiological doses) conditions [52]. When administered pharmacologically in the form of combined estrogen/progesterone contraceptive pills, estrogen does not inhibit fibroid growth, and its effect on fibroid-related heavy menstrual bleeding remains controversial [53]. On the other hand, pharmacological agents that block estrogen’s activity, including selective estrogen receptor modulators (SERMs) or gonadotropin-releasing hormone receptor (GnRH) agonists or antagonists, not only diminish fibroid-related heavy menstrual bleeding but they also lead to a decrease in fibroid size [54,55]. Intracellularly, when estrogen activates ER⍺, the MAPK and PI3K/Akt pathways become activated, stimulating proliferation and inhibiting apoptosis in uterine smooth muscle cells [52]. We recently found that PEGylated polymeric nanoparticles loaded with 2-methoxyestradiol, injected intraperitoneally in a patient-derived xenograft fibroid mouse model, led to a significant shrinkage of fibroids after one month of treatment [56]. In this study, we found that the viability of human uterine leiomyoma (HuLM) cells was diminished in a dose-dependent fashion, with the lowest viability at 10 μM, and an in vivo volume of human fibroid tissue markedly diminished in the 2-ME group compared to control after 28 days of treatment. Intriguingly, we observed a decrease in progesterone receptor (PR) expression but no change in ER expression in fibroids of the 2-ME treated group compared to control. The reason why obesity-related elevations in estrogen in the form of estrone may lead to fibroid growth rather than shrinkage remains uncertain.

How could the physiology of obesity affect fibroid growth in the background of elevated circulating estrone? A condition characterized by high circulating estrone and obesity is polycystic ovary syndrome (PCOS), which is characterized by anovulation or oligo-ovulation, imaging evidence of polycystic ovaries, and hyperandrogenism. Chronically elevated circulating levels of estrone, as seen in conditions including obesity and PCOS, are associated with anovulation, as estrone interferes with the feedback of estradiol on the hypothalamic–pituitary–ovarian (HPO) axis, impairing normal cyclic hormonal fluctuations that characterize a physiological menstrual cycle [57,58]. PCOS exemplifies a chronic anovulatory disorder with many of the hallmark features of the metabolic syndrome, including obesity, insulin resistance, and dyslipidemia. Moreover, estrone levels in patients with PCOS correlate positively with luteinizing hormone (LH) as well as the total antral follicle count (TFC) and total ovarian volume (TOV), suggesting that estrone may play a role in the pathogenesis of ovulation suppression in this condition [58]. The relationship between PCOS and fibroid risk appears mixed in the literature. For instance, in an observational cohort study of 2249 patients, Huang et al. compared the prevalence of fibroids among patients with PCOS versus those with unexplained infertility and found significantly lower fibroid prevalence in the PCOS group after controlling for covariates [59]. On the other hand, Wise et al. reported on a larger cohort of 114,000 person-years and found a positive association between PCOS and fibroids after controlling for covariates [60]. Thus, the relationship between fibroids and PCOS is likely complex and multifactorial, and it may depend on variables such as concentrations of circulating hormone levels and PCOS phenotype (lean versus obese).

The estradiol-dependent progesterone receptor regulation we observed in fibroid cells suggests that circulating estrogen levels directly impact fibroid signaling, leading to proliferation versus quiescence or apoptosis [56]. Progesterone receptor levels have also been linked to adipogenesis and obesity progression [61]. Furuhata et al. investigated the progesterone receptor membrane-associated component 1 (PGRMC1), a heme-binding membrane protein important in cell cycling and proliferation, in relation to a potential role in lipid accumulation in adipocytes. These investigators found that PGRMC1 is necessary for lipids accumulation in an embryonic fibroblast cell line [61]. Furthermore, PGRMC1 signaling leads to low-density lipoprotein (LDL) and very-low-density lipoprotein (VLDL) uptake via regulation of the LDL and VLDL receptors (LDL-R and VLDL-R). Thus, it stands to reason that estrogen-related progesterone receptor regulation could affect not only lipid accumulation and fat distribution in the body but potentially also key interactions between adipocytes and myometrial cells or MSCs. If, for instance, the obese state induced, through various estrogen-dependent or estrogen-independent pathways, increased PR signaling, this could contribute to aberrant adipocyte-MSC signaling, which could contribute to fibroid pathogenesis.

Here, we investigated the link between adipocytes and MSCs, hypothesizing that adipocyte-derived cues signal directly to stem cells in the uterine myometrium in a pro-inflammatory, high ROS microenvironment to promote DNA damage, mutation, and the genesis of uterine fibroids. The current findings suggest such a mechanism may be at least partially responsible for fibroid formation in the obese state. In particular, we found that adipocytes increase the proliferation of MSCs, increase ROS generation and trigger a histone-mediated DNA damage response pathway, likely culminating in the incitement of fibroid formation and propagation. Our findings provide a further line of evidence suggesting that cardiometabolic risk factors including obesity likely play a causal role in fibroid pathogenesis. 

The mechanisms underlying fibroid growth and development remain incompletely understood. Several germline genetic mutations have been strongly associated with risk of fibroid development during the lifespan. These include *MED12*, *HMGA2*, *fumarate hydratase*, and *COL4A5-COL4A6* gene mutations [62]. Intriguingly, oxidative stress associated with obesity may preferentially mutate *MED12* via NOX4 and TGF-β3-dependent pathways [8,9]. *MED12* codes for a key transcriptional regulator of RNA polymerase 2, and mutations of this gene lead to a highly penetrant hereditary form of leiomyomatosis [62,63]. The current study findings implicate *MED12* (although not exclusively this gene, as others may also be implicated) as potentially important in obesity-mediated fibroid genesis via a ROS–DNA damage pathway. Why certain fibroid-associated genes may be particularly prone to mutation in the setting of the pro-inflammatory, high ROS environment of obesity as compared to other gene loci remains to be determined. 

The role of miRNA in fibroid pathogenesis has been an intriguing development in the field of fibroid biology. Di Vincenzo et al. compared 15 miRNAs from healthy MSCs and fibroid-derived MSCs [64]. Using next-generation sequencing analysis, these investigators found the differential expression of miRNAs involved in a variety of cellular processes, including adherens junction formation, ECM signaling, and cell cycling. These differences in miRNA expression at the level of MSCs point to one ontological origin of the disease. Recently, Sprenkle et al. identified the contribution of an adipose tissue macrophage (ATM)-derived miRNA, miR-23, in regulating ATM proliferation through its effect on Eif4ebp2, using RNA sequencing [65]. This miRNA likely provides a link between the immune system and homeostatic regulation of energy balance. Whether and how these and other miRNAs differ between MSCs from obese versus normal weight people (with or without fibroids) remains to be studied. The present study begins to address how adiposity can influence MSC signaling and physiology. 

Limitations of the current study include the in vitro nature of the experiments conducted herein. It will be important to determine whether the interaction between adipocytes and MSCs observed here in the in vitro condition are recapitulated in vivo. Technically, this may be complicated to achieve, as cells and cell types in the living organism do not operate independently and are influenced by the systemic and hormonal milieu. Nevertheless, whether the signaling observed between adipocytes and MSCs in culture conditions may be altered by additional hormonal influences remains to be tested. Future investigations should seek to determine the conditions under which obesity contributes to fibroid development or growth in vivo. For instance, under conditions of PCOS, with central adiposity, insulin resistance, and hyperandrogenism, fibroid growth has been difficult to predict [59,60]. The growth of fibroids in the obese state likely depends on the complex interplay between cytokines, chemokines, and other hormonal factors circulating in the uterine environment.

In summary, we have demonstrated that adipocytes communicate directly with stems cells in the uterine myometrium, which express leptin receptors and upregulate the expression of such receptors under the influence of leptin and adiponectin, ultimately resulting in Wnt/β-catenin and TGF-β/SMAD-mediated MSC proliferation and collagen production. Whether these mechanisms fully capture the in vivo condition remains to be further elucidated. Clinical implications of the present study include the possibility that obesity or excess adiposity may be one target for fibroid treatment. An interesting possibility that remains to be studied is whether weight loss may be an effective strategy from the medical treatment of fibroids among obese individuals. Furthermore, a novel medical treatment of fibroids could be to disrupt inflammatory signaling through the inhibition of NOX4 or leptin receptors.

## Figures and Tables

**Figure 1 genes-14-01625-f001:**
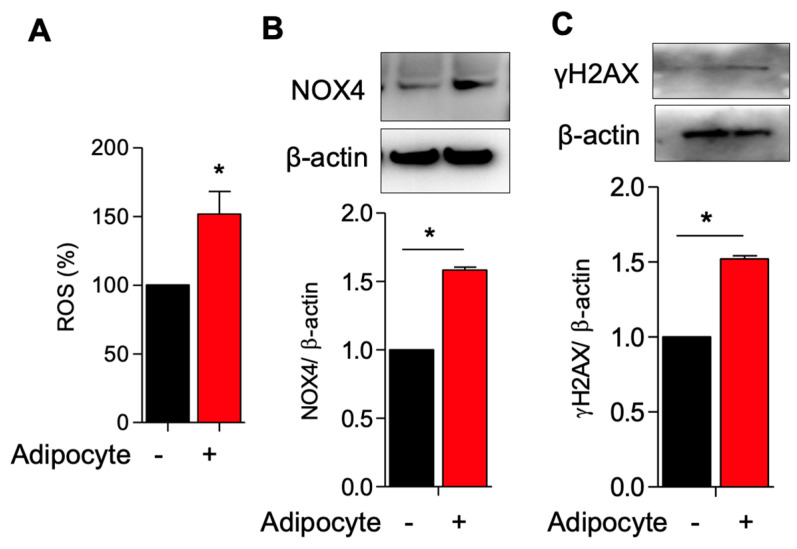
Adipocyte induces ROS production and activates NOX4 and γH2AX expression in myometrium stem cells co-culture. Human adipocyte (SW872) cells were co-cultured with myometrium stem cells for 8 days. (**A**) Changes in intracellular ROS levels were determined using the fluorogenic probe 5-(and-6)-chloromethyl-2′,7′-dichlorodihydrofluorescein diacetate acetyl ester. (**B**) NOX4 and (**C**) γH2AX protein expression were measured after 8 days co-cultured with adipocyte. Β-actin was used as a loading control. Data are presented as the means ± SEM of the relative expression obtained from three independent experiments. *, *p* < 0.05. Black bars denote MSC culture alone, red bars denote adipocyte-MSC co-culture.

**Figure 2 genes-14-01625-f002:**
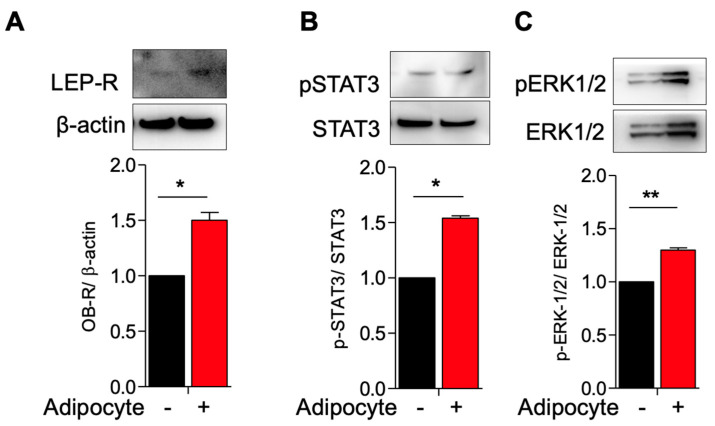
Activation of leptin receptor in myometrium stem cells co-culture with adipocyte. Human adipocyte (SW872) cells were co-cultured with myometrium stem cells for 8 days. (**A**) LEP-R, (**B**) pSTAT3/STAT3 and (**C**) pERK1/2/ERK1/2 protein expression were measured after 8 days co-cultured with adipocyte. Β-actin was used as a loading control. Data are presented as the means ± SEM of the relative expression obtained from three independent experiments. *, *p* < 0.05. **, *p* < 0.01. Black bars denote MSC culture alone, red bars denote adipocyte−MSC co-culture.

**Figure 3 genes-14-01625-f003:**
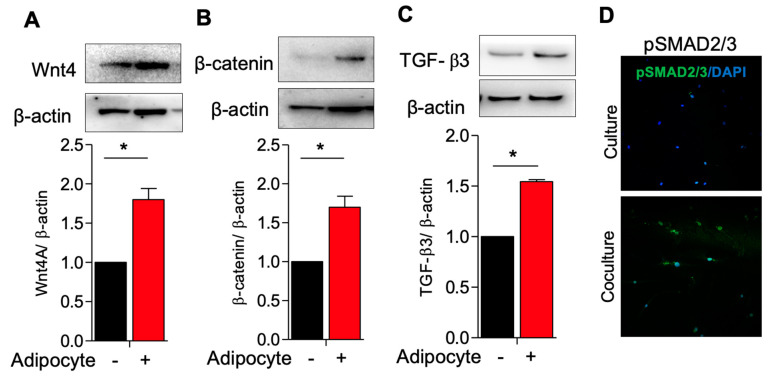
Activation of TGF-B3/SMAD2 and Wnt4/β−catenin pathway in myometrium stem cells co-culture with adipocyte. Human adipocyte (SW872) cells were co-cultured with myometrium stem cells for 8 days. (**A**) Wnt4, (**B**) β-catenin and (**C**) TGF-β3 protein expression were measured after 8 days co-cultured with adipocyte. (**D**) Β-actin was used as a loading control. Immunofluorescence staining was performed on myometrium stem cells after 8 days co-cultured with an antibody against pSMAD2/3 (green) and nuclei were stained with DAPI (blue). All images were captured with the same time exposure using a confocal microscope (20× magnification). Data are presented as the means ± SEM of the relative expression obtained from three independent experiments. *, *p* < 0.05. Black bars denote MSC culture alone, red bars denote adipocyte−MSC co-culture.

**Figure 4 genes-14-01625-f004:**
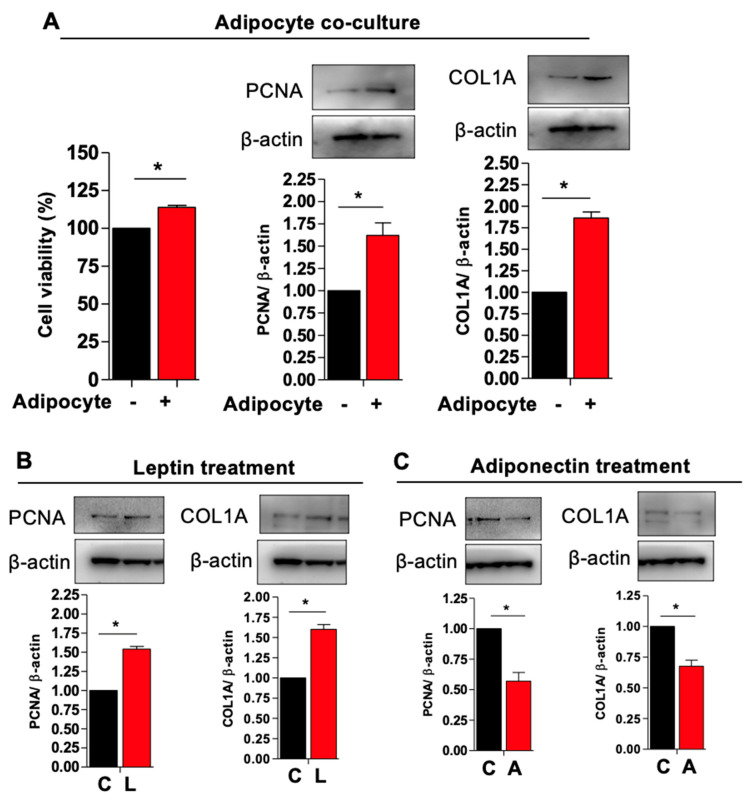
Adipocyte co-cultures and adipokine treatment increases myometrium stem cells proliferations and collagen type 1 production. Human adipocyte (SW872) cells were co-cultured with myometrium stem cells for 8 days. (**A**) Viability was assessed by (MTT) staining by measuring absorbance at 500–600 nm. (**A**) PCNA and COL1A protein expression were measured after 8 days co-cultured with adipocyte. Β-actin was used as a loading control. For the leptin and adiponectin treatment, cells were treated with 100 ng/mL leptin and 20 μg/mL adiponectin for 48 h. (**B**,**C**) After 48 h of incubation, cells were harvested for PCNA and COL1A protein expression. Β-actin was used as a loading control. Data are presented as the means ± SEM of the relative expression obtained from three independent experiments. *, *p* < 0.05. Black bars denote MSC culture alone, red bars denote adipocyte−MSC co-culture.

**Figure 5 genes-14-01625-f005:**
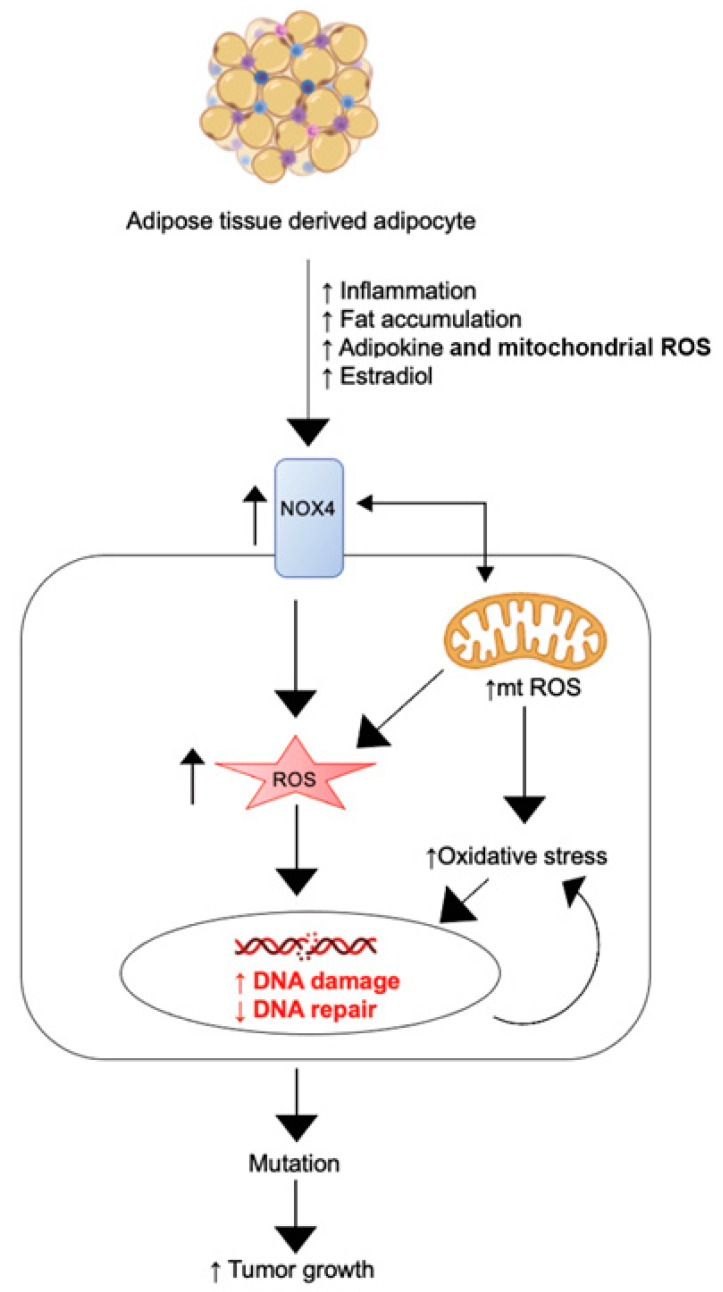
Adipocyte-induced signaling to myometrial stem cells promotes the generation of ROS, DNA damage, and subsequent fibroid generation.

## Data Availability

Data are available upon request to the corresponding author.

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
