# Peer review of "Obesity Contributes to Transformation of Myometrial Stem-Cell Niche to Leiomyoma via Inducing Oxidative Stress, DNA Damage, Proliferation, and Extracellular Matrix Deposition"

_genes, 2023, doi:10.3390/genes14081625_

Round 1
Reviewer 1 Report
The article used in vitro system to study potential roles of obesity in uterine leiomyoma tumorigenesis. The topic and data are very interesting.
My suggestions:
1. Abstract: the following abbreviations were not defined at first use.
Line 6 MSCs
Line 7 FACs ELISA
Line 10 ECM
Line 11 MTT
2. Page2 line 10 and 11 fonts are inconsistent.
3. Page2 Line 13 “We hypothe-sized that adipokines myometrial stem cells (MSC), the precursor cells of fibroids, target-ing inflammatory pathways. “ Something is missing in this sentence.
4. ROS was never defined.
5. CLARIOstar needs company name and state name.
6. In the discussion, maybe author could discuss what they think estrogen, which is over produced by excess adipocytes could potentially fit in this picture.
7. In the discussion, maybe author could discuss what are the limitations of this study, for example, how cell culture vs. native in vivo environment, etc.
Minor editing of English language required
Author Response
- Abstract: the following abbreviations were not defined at first use.
Line 6 MSCs
Line 7 FACs ELISA
Line 10 ECM
Line 11 MTT
We have clarified these abbreviations in the abstract
- Page2 line 10 and 11 fonts are inconsistent.
We have made the fonts consistent.
- Page2 Line 13 “We hypothe-sized that adipokines myometrial stem cells (MSC), the precursor cells of fibroids, target-ing inflammatory pathways. “ Something is missing in this sentence.
Thank you for pointing this out. We have revised this sentence.
- ROS was never defined (page 6).
We have defined this term.
- CLARIOstar needs company name and state name.
We have added this information.
- In the discussion, maybe author could discuss what they think estrogen, which is over produced by excess adipocytes could potentially fit in this picture.
We thank the reviewer for this suggestion. We have added to our discussion (page 10).
- In the discussion, maybe author could discuss what are the limitations of this study, for example, how cell culture vs. native in vivo environment, etc.
We have added a section on limitations to our study in the discussion.
Reviewer 2 Report
I am satisfied that authors after their wide and well planned work did not summarize that obesity causes leiomyoma but gave only suggestion that obesity may mediate MSC initation of tumorigenesis resulting in myoma, And it is exactly conclusion of their work
Author Response
We thank the reviewer for their positive comments on our manuscript.
Reviewer 3 Report
The authors have done a good job trying to elucidate the mechanism through which adipocytes and their receptors promote leiomyomas. Oxidative phosphorylation pathways are always over expressed in cancer cells and the signaling pathways are immensely crosslinked. But the manuscript provides convincing evidence of STAT3 and Erk12 pathways involvement. In the methods section, the authors state that the experiment was done five times. Was this only for the ROS production assay or even for the cell culture and western blotting? If the authors can clarify this on the paper, there will be no doubt on the reproducibility of the results.
Minor changes are required in the introduction section. Different fonts are used.
Reviewer 4 Report
This study investigated the influence on adipocytes on myometrial stem cells behaviour. It is an interesting manuscript. I have no comments regarding the methods and results sections. Is there a possitive association between BMI and the size of myomas? This should be added to the introduction section. In addition, in the conclusions section the authors should further discuss a possible clinical implications of this study.
Author Response
The reviewer raises an interesting question pertaining to a possible dose-responsiveness between adipose volume and fibroid size. We have included a study that found a positive association between BMI and fibroid size (page 3). Additionally, we have added the clinical implications of our study in our discussion (page 11).